# Adhesive Bonding of Scots Pine Wood from the Szczecinek Forest District for the Production of Garden Equipment: An Experimental Investigation

**DOI:** 10.3390/ma15248994

**Published:** 2022-12-16

**Authors:** Tomasz Krystofiak, Barbara Lis, Cezary Beker

**Affiliations:** 1Department of Wood Science and Thermal Techniques, Faculty of Forestry and Wood Technology, Poznan University of Life Sciences, 60-637 Poznan, Poland; 2Department of Forest Management, Faculty of Forestry and Wood Technology, Poznan University of Life Sciences, 60-625 Poznan, Poland

**Keywords:** pine wood, PUR adhesive, protection agent, strength, durability, aging, point scale

## Abstract

This work aimed to determine the gluability of pressure-impregnated pine wood with two protection agents used in production technologies for garden equipment and their effect on the strength, water resistance and thermal resistance of joints, as well as their susceptibility to aging. The tests were carried out on Scots pine wood (*Pinus sylvestris* L.) purchased from the forest districts of the Regional Directorate of State Forests in Szczecinek town, Poland. For the impregnation, two commercial protection agents were used. The pressure impregnation of the elements and gluing with the PUR adhesive were carried out by a garden furniture manufacturer. The strength and durability after aging tests of glued joints were performed in accordance with the procedure described in Technical Approval No. AT-15-2948/00 and the PN-EN ISO 9142 standard. The collected experimental data can be used in technological solutions, mainly for processes which involve gluing wood after impregnation.

## 1. Introduction

Wood continues to compete successfully with other materials used in various branches of industry. When considering the ecological balance, wood-based products provide many benefits in terms of their sustainable development and physico-mechanical properties. Conifer wood, particularly pine, is widely used in the construction industry to make a variety of engineering structures, finishing elements and fittings, including those used in garden architecture. However, when used in outdoor conditions, wood needs to be suitably protected; otherwise, changes may occur, leading to its technical degradation. The degree of such changes depends, above all, on the species and the features of its anatomical structure. The protection of functional wood, particularly that coming into direct contact with water or soil, is largely based on introducing protection agents with powerful, conserving, fungicidal properties. The range of such products is constantly being expanded with the introduction of new generations of active substances, which are currently being developed with regard to eco-safety requirements [1,2,3,4,5]. Generally speaking, wood impregnation processes are carried out by surface methods (lubrication, spraying and immersion) or vacuum pressure methods. They must be implemented with rigorous technological accuracy. An inappropriate selection of parameters, such as the concentration of the impregnating solution, the humidity of the wood, pressure or process time, will have an adverse impact on the effectiveness of impregnation and, thus, also on the durability of the final products.

In subsequent technological operations, protected wood is often subjected to gluing processes. Glued laminated wood, suitably protected with biocidal substances, offers high functionality and durability. For glued wooden elements exposed to atmospheric conditions, they should be held together with bonding substances that ensure the highest strength against various static and dynamic loads, resistance to the effects of water and temperature, and while it’s used to fasten wood to protected surfaces—non-sensitivity to the protection agents [6,7,8]. In the woodworking industry, wood production for gardens is a fairly wide field and is very popular in Poland and Western European countries. The technology used for production and the resulting strength and resistance of glued elements in such products greatly influence their final quality.

Binding substances of interest, preferred (among other things) for the gluing of impregnated wood, include PUR adhesives, which are available in both one- and two-component versions. In view of the variety of ways in which they can be used and the possibilities of crosslinking, they constitute a group of high-quality products with multifaceted applications. They offer excellent adhesion to many types of bases, relatively quick hardening, elasticity and the formation of joints with good thermal resistance, particularly at negative temperatures [9,10,11,12,13]. Adhesion technologies with PUR systems can be used for gluing wood with high moisture contents, even as high as 25%. The one-component PUR adhesives are prepared from compounds with an active hydrogen atom mixed in excess with isocyanate prepolymers, which, as a result, the macromolecules of the resulting polyurethane contain free NCO groups. The crosslinking of these adhesives results from a reaction of the NCO groups through contact with water contained in the air or in the material being glued. The carbamic acid that forms during the reaction is unstable and decomposes, splitting off carbon dioxide to form an amine, which, in turn, reacts with other isocyanate groups, forming polyurethanes [14,15].

One solution often applied in industrial practice is the impregnation of pre-glued elements as one of the final operations in the production cycle. However, this method requires certain conditions to be met, primarily concerning the methods of impregnation. The treatment of glued elements is usually limited to the simplest methods of surface protection. In the case of deep impregnation, when it is necessary to use the vacuum pressure methods, numerous difficulties arise, including a swelling of wood, moisture-induced stress, increased roughness of the protected surfaces, problems with the resistance of glued joints in the conditions of the impregnation process, and the need for repeated drying of the impregnated elements. Significant limitations often arise from the dimensions of autoclaves, which permit the impregnation of glued semi-finished products, but only if they are of relatively small size. Because of these reasons gluing elements that have been previously impregnated gives bigger possibilities. Solutions of this type ensure the resistance of glued systems on their entire cross-section, which is achieved much more effectively in the case of gluing easily impregnable pine wood. This approach also has other technical and economic advantages [16,17,18,19,20,21]. The substances impregnated into the wood may change the physicochemical state of the surface and thus affect the phenomena of wetting, spreading and the penetration of adhesives during the bonding process. As a result of the dissolution or diffusion, they may penetrate into the adhesives, exerting a catalytic or inhibiting effect on the solidification processes and other features of importance during the application, such as the open assembly time, gelation time and pressing time, as well as the degree of hardening of glue in the joint [3,18,22,23,24,25,26,27]. In evaluating the effect of the type of wood protection agent in interaction with glues, an important factor is the composition of this substance, and in particular the presence of metals in ionic form. These problems have also been considered in publications in relation to the processes of adhesion and hardening [26,28,29,30]. A number of works have been dedicated to technological questions, in relation to both solid and engineered wood, with an analysis of the significance of the species of wood, the type of protection agent and method of application, and the type of adhesive and pressing parameters. The results of these studies indicate a varied influence of protection agents on the strength of joints. In most cases, the comparisons show that such substances cause deterioration in the physico-mechanical properties of the wood and the strength and resistance of joints [3,7,31,32,33,34,35,36,37,38,39]. However, some investigations have found that protection agents do not have an adverse impact on the gluing process and may even allow for a slightly higher strength to be obtained [40,41].

In summary, as shown by a review of data from the literature concerning the suitability of impregnated wood for gluing, these studies cover a wide spectrum of questions. A systematic evaluation of the results obtained by different researchers is made especially difficult by the extremely divergent methodological approaches used, both in terms of the experimental methods and the criteria used for evaluation.

In industrial practice, it is important to determine the functional parameters before the products are launched in the market. In view of the variety of impregnating and binding substances in use, in the course of the technological process, every case of gluing requires individual analysis [42]. In the design of glued joints, it is important to know about the relevant mechanical properties. It means that it is necessary to perform strength tests on such joints. Such testing is particularly essential because, as noted above, the variability of properties of wood after impregnation often has a negative impact on strength. The range of tests carried out should be appropriate to the intended function of the product and should take into account the conditions in which it will be used. Most commonly performed are the accelerated laboratory tests using standardized methods, which allow the verification of the interaction between wood and adhesives under appropriate environmental conditions [8,43].

In this context, the need has arisen for research to investigate and quantify the gluability of the pine wood (*Pinus sylvestris* L.) popular in the Polish timber industry, having a treated surface and which is intended for use in elements of small garden architecture. This work aimed to determine the gluability of pressure-impregnated pine wood with two substances used in production technologies for garden equipment and their effect on the strength, water resistance and thermal resistance of joints, as well as their susceptibility to aging. The collected experimental data may be used in technological solutions, mainly for processes which involve gluing wood after impregnation.

## 2. Materials and Methods

Tests were carried out on Scots pine wood (*Pinus sylvestris* L.) purchased from the forest districts of the Regional Directorate of State Forests in Szczecinek, class WB0-1 (large-dimension timber, thickness class 1). For the impregnation, two commercial protection agents were used: X (a liquid, chromium-free wood preservative salt based on inorganic copper, boron compounds and organic ingredients) and Y (a liquid salt concentrate based on copper and chromium compounds; soluble in water). The impregnation of the elements (length 1930 mm, thickness 9 mm) in the form of slats was carried out by a pressure method, subject to the technological conditions applied by a manufacturer of garden furniture, using two commercial substances based on the compounds of copper and boron (Table 1).

The impregnated wood was glued using a one-component commercial PUR adhesive (with a density at 25 °C 1.1–1.2 g/cm^3^, viscosity at 25 °C 7500–8500 mPa·s, pressing pressure of 0.1–0.5 MPa, open assembly time of 5–10 min and a pressing time of up to 60 min).

Prior to application, the adhesive was heated to approximately 40 °C, and it was then applied to the suitably prepared slats using a roller (200 ± 20 g/m^2^). Next, sets were formed and placed in a pneumatic mold, where they were pressed under a minimum pressure of 1.5 MPa for a time of 1.5 h. Figure 1 shows a view of the arches prepared under production conditions. From the resulting finished arched elements, block samples of the dimensions 50 × 50 × 40 mm were prepared [44]. Figure 2 shows a block sample (a schematic diagram and its photo) for testing the strength and resistance of the glue lines, along with the way along with the method of loading. Tests were carried out on both unprotected and impregnated wood.

Samples were subjected to the tests defined in the technical approval issued by the Institute of Building Technology (ITB) in Warsaw, used for the evaluation of the quality of joints in single-frame windows and balcony doors made from glued laminated wood [45]. The test conditions and requirements for particular joint resistance classes are provided in Table 2.

The aging tests of glued joints were performed in accordance with the procedure described in PN-EN ISO 9142 for the tests denoted D7 and D11, which simulate the effect of atmospheric conditions and are applicable to glued wooden elements used outdoors [46]. The first test (D 7) included the following operations:64 h soaking in water at temp. (23 ± 2) °C.8 h drying in the temp. (55 ± 2) °C.16 h soaking in water at temp. (23 ± 2) °C.8 h drying in the temp. (55 ± 2) °C.16 h soaking in water at temp. (23 ± 2) °C.8 h drying in the temp. (55 ± 2) °C.16 h soaking in water at temp. (23 ± 2) °C.

The stages of the second aging test (D 11) were as follows:1 h in water at temp. (100 ± 2) °C.20 h drying in air at temp. (50 ± 2) °C.1 h in water at temp. (100 ± 2) °C.Cooling of samples immersed in water to a temp. (20 ± 1) °C.

The samples were subjected to aging in three cycles, according to the above procedures, and the strength of the joints was determined after each cycle.

The tests of joint strength were performed using a Schopper ZDM 2.5/91 strength testing machine (Veb Thüringer Industriewerk, Germany, Rauenstein) with the load increment rate selected according to the ITB procedure so as to obtain delamination of the joint at a time of 90 ±30 s from the placement of the sample in the clamps of the testing machine. Immediately after the strength tests, a visual inspection of the surface was made for each sample by recording the wood failure percentage (WFP) factor. The value of the WFP factor was estimated with an accuracy of 10%. This provided an additional criterion for the estimation of joint quality.

The results obtained were used for a comprehensive evaluation of the properties of joints made with the PUR adhesive, in curved elements for use in garden equipment, based on a proposed point scale. In this context, a system was developed that enabled the evaluation of the joints. Table 3 and Table 4 contain computational formulas and evaluation criteria in terms of the changes relative to test No. 1. The overall evaluation of the quality of joints is given by the sum of the individual formula.

## 3. Results

The individual tests were performed for 16 samples. The results of the joint strength and resistance tests are provided in Table 5, Table 6 and Table 7, which contain values of X_avg._, x_max._, x_min._, υ.

## 4. Discussion

The analysis of the average values of the immediate strength of the joints (Table 5—test 1) showed that the glued joints in the tested systems produced favorable results in terms of both the values obtained and images of delamination (WFP). The shear strength values lie within a range of 8.22–8.83 MPa, and it should be noted that the WFP value, at destructive loads, was 100%. This demonstrates that the glued wood was the weakest area of the tested joints. According to data in the literature, it is related to the fact that both the force of cohesion between the glue molecules and the force of adhesion to the base were dominant over the forces of cohesion in the wood. For this reason, the values presented in the results reflect the shear strength of the wood and not of glued joints [47]. The protection agents did not mark any negative impact on the gluing process or the properties of joints.

In individual tests (Table 5—tests 2 and 3), the strength and resistance of joints under the action of water, as defined in the ITB approval, were found to decrease. When taking into account the conditions applied in the test procedures, the differences in the results between variants were found to be insignificant. Generally, the largest decrease in joint strength was recorded following test no. 2. This decrease was approximately 64% for the joints made with unimpregnated wood and 54–56% for protection wood (Y and X). It should be noted that, for most of the tested samples, wood delamination has occurred.

The fall in the mechanical strength of the tested variants that followed test no. 3 may be ascribed to the processes of sorption and desorption. This control, among other things like—important characteristics of the wood tissue-shrinkage swelling and formation of thermal and moisture stresses—has an impact on the mechanical performance. Changes in shape and dimensions lead to the delamination of wood or glued joints and, thus, can cause a reduction in durability [8].

The results obtained in this study are in agreement with those of other researchers, who have reported a reduction in the water resistance of tested samples immediately after exposure to the effect of cold water, or in combination with a conditioning process [42,43,48,49,50]. Analyses of the strength of joints compared with the maximum values showed, unambiguously for all systems, that the PUR adhesive used, when applied both to impregnated and unprotected wood, provides the technical possibility of obtaining water-resistant glued joints.

In turn, the strength of the joints following test no. 6 (Table 5), which are for the different variants involving the gluing of both unprotected and impregnated wood, lies within the range of 7.20–8.74 MPa.

A comparative analysis of the results showed that the substances used for wood impregnation caused a reduction in the thermal resistance of joints by 8–18%. The greatest changes concerning the control joints made from unprotected wood were noted for the joints obtained by gluing pine wood protection with substance X.

Despite a slight decrease in strength when compared to the control joints, the tested systems still exceeded the value laid down as a criterion for this test. It should also be clearly highlighted that in the evaluation of the images of the delamination of joints observed, followed by the test in which samples were heated, the dominant delamination was in the wood zones (WFP in the range of 90–100%), which is an additional argument confirming the good thermal resistance of the joints.

It was found in a comprehensive analysis of the aging test results (Table 6 and Table 7) that, as a result of the operations performed cyclically under the conditions set out in the relevant procedures, the joint strength was reduced compared to the control samples, even after the first cycles were completed. When the changes in strength caused by the aging processes were expressed in relative terms, as the ratio of the strengths of joints made with impregnated and unprotected wood (immediate strength—Table 5—test 1), after a defined number of cycles, it was found that the protection agents had a similar effect on these values. A decrease in strength by approximately 40% was recorded for the joints made with the PUR adhesive after three aging cycles in accordance with PN-EN 29142 (Table 6 and Table 7). When comparing the changes in joint strength after a defined number of cycles, the effect of the aging tests was found to be fairly strongly differentiated. In the conditions of the procedure aging test D7, including the cyclic immersion of samples in water and drying, it was found that the strength of the glued joints decreased more slowly than in the case of the second test described in PN-EN 29142 (aging test D 11). The strength decay after three aging cycles was approximately 40%, while in the case of the D11 aging procedure, such a level was reached only after one cycle. When taking account of the further dynamics of reduction in the strength of joints relative to the control samples, it was found that joints that were considerably weakened at the start of the test generally stabilized in strength at subsequent stages, while in the case of the systems weathered in accordance with the requirements of test D7, the resistance decreased gradually.

The literature data indicate that cyclic changes relating to the sorption and desorption processes are responsible for a significant degree of deterioration in the strength parameters of glued joints. A large reduction in strength may be recorded, even in the first period of cyclic loading [31,33,48,51,52].

Analysis of the WFP indicator after the aging cycles showed that the joints, which were made with the use of the tested adhesives, were characterized by similar resistance to aging conditions procedures specified in PN-EN 29142. For most systems, regardless of the number of cycles, the values of that parameter lie within a range of 80–100%. This confirms that the joints obtained have a high resistance to aging and that, decidedly, the weakest link in all the variants considered was glued wood.

The coefficients of variation computed for the presented experimental data covered a wide range: 9.3–21.7% for the tests of “conditioned” samples (Table 5—tests 1, 2 and 3) and 6.9–11.7% for the test of thermal resistance (Table 5—test 6). The values, as functions of the number of aging cycles carried out according to D7 (procedures described in Table 6), were in the range of 5.3–15.2%, and those for PN-EN 29142 (D11 aging test—Table 7) were in the range of 5.1–11.9%. High values of ν correspond to the data provided in the literature for tests of this type and are accepted as fully reliable, thus forming a basis for inference [42,53]. It may be noted that the experimental data obtained for the joints made by gluing impregnated wood, particularly with the X preservation agent, gave, in comparative terms, the values of ν that were slightly higher (by several percent), which indicates a larger scatter of measured values.

In order to provide a complementary assessment of the effect of the impregnating substances used in the study on the gluability of the treated pine wood in terms of the strength and resistance of the glued joints, the experimental data have been considered in light of the proposed evaluation criteria provided in Table 8. The boundary value enabling the formulation of a positive evaluation and acceptance of a given variant as “suitable for gluing” was assumed to be 60% of the total available points in the adopted evaluation formula. If a variant failed to reach the given value, it was classified as having “limited suitability for gluing”, meaning, in other words, that the given wood—protection agent—adhesive system enables the obtaining of joints of lower resistance, thus limiting the range of potential applications for that system of glued wood.

The results obtained following ITB technical approval test No. 1 were used as a control (Table 5). Table 8 provides the total point scores.

From an analysis of the data in Table 8, it was concluded that the PUR adhesive is suitable for the gluing of impregnated curved elements for use in garden equipment. The systems with the protection agent scored 38 points, which is approximately 85% of the total available.

In order to supplement the data and provide an even fuller evaluation of the influence of the wood protection agent on the quality of glued joints, a further criterion was applied, requiring an analysis of the relative strength of joints, expressed as a ratio of the value for glued impregnated pine wood to the value for unprotected wood. Table 9 provides the results of these calculations in terms of the point scores for joints after the individual resistance tests.

On the adopted scoring scale, the tested systems could obtain a maximum of 20 points in total. In the case of the pine wood impregnated, respectively, with the protection agents Y and X, the results of the computations for the joints were 17 points, which indicates that the wood protected using these substances is suitable for gluing. All the tested systems obtained the maximum point scores following that aging procedure, which proved to be significantly milder than the D7 procedure. Additionally regarding the D7 test we observed a gradual decrease in strength as a function of completed aging tests.

## 5. Summary

Based on the results of the tests and the assumption criteria adopted for their evaluation, it is concluded that a PUR adhesive can be successfully used for the gluing of pine wood impregnated with the tested substances, thus obtaining joints with high resistance to the effects of water, heat and aging.

In the evaluation of the mechanisms of the delamination of joints at destructive loads in the tests of water resistance, the significance of the WFP value turned out to be very high. Regarding the joint resistance tests carried out with the conditioning of samples, for the great majority of variants, the parameter exceeded 90%. This indicates that the presented test results of the strength and resistance of joints do not express the technical possibilities of the adhesives in a fully adequate manner and, to a large degree, refer to the specific properties of the glued wood, both unprotected and impregnated. It should be noted that favorable values of the WFP were also observed by the aging tests of joints. Practically, except in a few cases, the parameter attained a level of 100%.

The tests in this study represent the first stage in an evaluation of pine wood from Szczecinek Forest District for producing elements made of laminated glued wood that have been previously impregnated for use in areas such as garden architecture. Additionally, we plan to carry out further experiments, in this regard, on wood originating from the stands growing in forest areas and former agricultural land. We will pay attention to the potential disadvantages of the initial treatment (proper wetting and adhesion of the glue to the protected surface) of the use of additional technological operations before gluing, such as removal of crystallized preservatives on the surface of the glued wood.

## 6. Conclusions

The following conclusions were drawn from the tests:The PUR adhesive that was used in the tests is suitable for gluing Scots pine (*Pinus sylvestris* L.) wood, which was pressure-impregnated with two substances used in production technologies for garden equipment.In the case of the impregnated wood, the PUR glue joints met the criterion requirements for water and heat resistance, according to the technical approval of the Institute of Building Technology AT 15-2948/00.The obtained adhesive joints showed, both in terms of the obtained values and the images of delamination expressed by the WFP factor, very favorable relations, indicating their high quality. The weakest zone in the evaluated joints, determining the level of the obtained values, was wood.The adopted point scoring system was found to be useful as a criterion for the gluing quality, enabling a classification of the joints on the basis of the overall analysis, including water, thermal resistance and aging tests.In the point-based analysis, the systems with impregnated wood obtained 38 points, approximately 85% of the total available.

## Figures and Tables

**Figure 1 materials-15-08994-f001:**
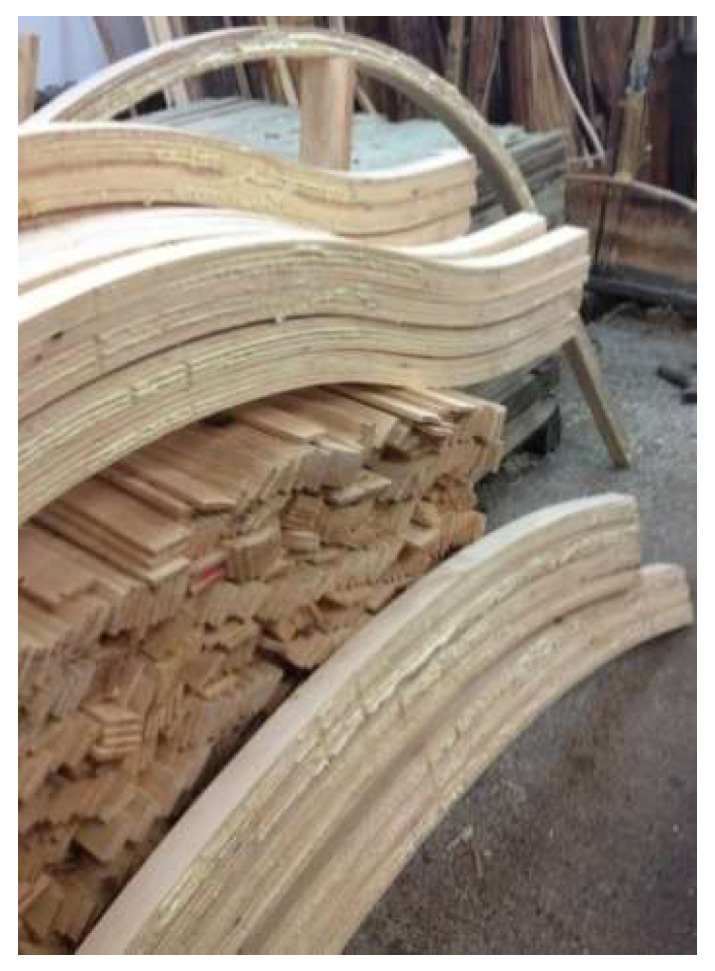
Arches prepared under production conditions.

**Figure 2 materials-15-08994-f002:**
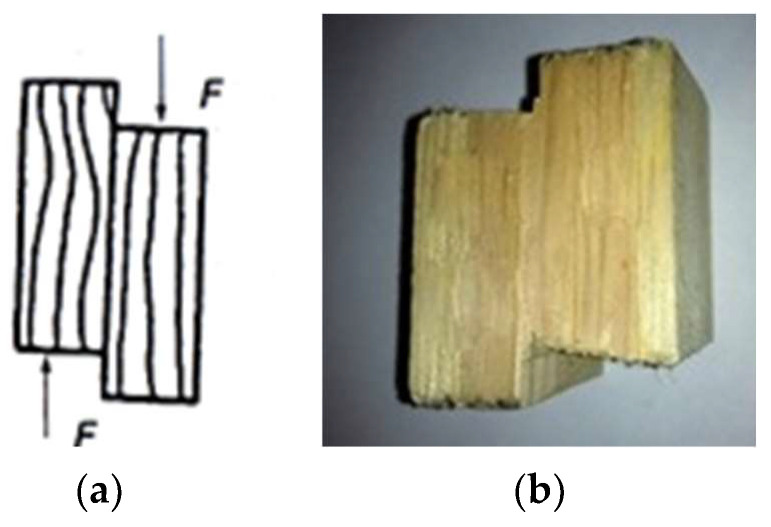
Block sample for testing the strength and resistance of glue lines: (**a**) schematic diagram; (**b**) photo.

**Table 1 materials-15-08994-t001:** Basic properties of protection agents used for investigation.

**Properties**	**Protection Agent X**	**Protection Agent Y**
Density in the temp. 23 °C [g/cm^3^]	1.20	1.65
pH value	ca. 9.6	ca. 2–3 (4% solution)
Viscosity in the temp. 20 °C [mPa·s]	100 mPa·s	-

**Table 2 materials-15-08994-t002:** Strength and resistance tests of adhesive joints according to the ITB technical approvals AT-15-2948/00 [45].

No. of Tests	Tests Conditions	Shearing Strength [MPa]
1	7 ^1)^ days in the normal climate ^2)^	≥9.0
2	7 days in the normal climate 4 days in cold water ^3)^	≥3.5
3	7 days in the normal climate 4 days in cold water 7 days in the normal climate	≥7.0
4	7 days in the normal climate 6 h in boiling water 2 h in cold water	≥2.5 (not carried out)
5	7 days in the normal climate 6 h in boiling water 2 h in cold water 7 days in the normal climate	≥7.0 (not carried out)
6	7 days in the normal climate 3 h in the temp. 80 (±2) °C (Examination in terms of procedure WATT 91)	≥5.5

^1)^ day—24 h. ^2)^ normal climate—temp. 20 (±2) °C, relative humidity 65 (±5)%. ^3)^ water with temp. 20 (±2) °C.

**Table 3 materials-15-08994-t003:** Rating scale based on the relative strength and resistance of glue lines.

Calculation Formula	Assumptions Criterion	Scale (Degree)
I. Glue lines water resistance test (“in wet state”) TEST 2, acc. to the Institute of Building Technology procedure test 2 R_2/1_ = ---------------------------------------------- × 100 [%] test 1	≥80	5
	≥60	4
	≥40	3
	≥20	2
	<20	1
II. Testing of the glue line water resistance (“after drying”) test no. 3, acc. to the ITB procedure test 3 R_3/1_ = ---------------------------------------------- × 100 [%] test 1	≥80	5
	≥60	4
	≥40	3
	≥20	2
	<20	1
III. Testing of the thermo-resistance of glue lines (80 °C/3 h) test no. 6, acc. to the ITB procedure test 6 R_6/1_ = ---------------------------------------------- × 100 [%] test 1	≥80	5
	≥60	4
	≥40	3
	≥20	2
	≤20	1
IV. Aging tests D7 and D11, acc. to the PN-EN ISO 9142 standard Aging test D7 or D11 R_D7 lub D11_ = ------------------------------------- × 100 [%] Test no. 1 (acc. to the ITB procedure)	≥80	5
	≥60	4
	≥40	3
	≥20	2
	≤20	1

**Table 4 materials-15-08994-t004:** Scale for estimation of the resistance of glue lines after aging tests.

Calculation Formula	Assumptions Criterion [%]	Scale (Degree)
Aging tests D7 and D11, acc. to the PN-EN ISO 9142 standard (after individual cycles) Aging test D7 or D11 after the second and third cycles R_cD7 lub cD11_ = ---------------------------------------- × 100 [%] Aging test D7 or D11	≥90	5
	≥80	4
	≥70	3
	≥60	2
	<60	1

**Table 5 materials-15-08994-t005:** Results of investigation of water and thermal resistance of glue lines from pine wood joints after tests acc. to ITB procedure [45].

Test No. acc. to ITB Procedure (Table 2)	Statistical Data			
	X_min._	X_avg._	X_max._	ν
	[MPa]			[%]
	unprotected wood			
1	6.87	8.83	10.72	12.29
2	2.73	3.20	4.67	16.06
3	5.22	6.37	8.64	12.68
6	7.82	8.74	9.90	6.89
	wood protected with the X protection agent			
1	6.52	8.22	11.36	16.06
2	3.07	3.62	4.60	14.18
3	5.96	7.09	7.84	9.28
6	6.21	7.20	9.06	11.70
	wood protected with the Y protection agent			
1	5.65	8.52	12.17	21.72
2	3.22	3.91	4.82	11.33
3	5.81	7.12	8.32	10.96
6	6.54	8.11	9.12	11.62

**Table 6 materials-15-08994-t006:** Results of aging tests of glue lines from pine wood joints after D7 aging tests acc. to the PN-EN ISO 9142 standard [46].

Number of Cycles	Statistical Date			
	X_min._	X_avg._	X_max._	ν
	[MPa]			[%]
	unprotected wood			
1	5.25	6.42	8.61	15.21
2	4.90	5.83	7.39	10.65
3	4.22	5.13	5.93	9.35
	pine wood protected with X protection agent			
1	6.46	7.36	8.61	8.96
2	5.24	6.22	7.42	11.68
3	4.64	5.50	7.04	12.87
	pine wood protected with Y protection agent			
1	6.44	7.49	8.67	8.99
2	5.35	6.34	7.35	9.95
3	5.17	5.51	5.98	5.27

**Table 7 materials-15-08994-t007:** Results of aging tests of glue lines from pine wood joints after D11 aging tests acc. to the PN-EN ISO 9142 standard [46].

Number of Cycles	Statistical Data			
	X_min._	X_avg._	X_max._	ν
	[MPa]			[%]
	unprotected wood			
1	5.01	5.66	7.10	8.65
2	4.53	5.18	6.06	8.46
3	4.46	5.19	6.03	9.75
	pine wood protected with X protection agent			
1	4.62	5.51	7.31	11.69
2	4.44	5.33	6.14	10.85
3	4.55	5.45	6.71	11.94
	pine wood protected with Y protection agent			
1	4.87	5.59	6.37	8.33
2	4.87	5.28	5.74	5.12
3	5.03	5.58	6.65	8.21

**Table 8 materials-15-08994-t008:** Point scores for adhesive joints after individual tests (acc. to the criterion in Table 3).

Configurations	Test Type									Sum
	Test No. acc. to the ITB procedure			D7 i D11 acc. to the PN-EN ISO 9142 standard						
	2	3	6	D7-1	D7-2	D7-3	D11-1	D11-2	D11-3	
	Points (scale according to Table 4)									
Pine wood	2	4	5	4	4	3	4	3	3	32
Pine wood + X protection agent	3	5	5	5	4	4	4	4	4	38
Pine wood + Y protection agent	3	5	5	5	4	4	4	4	4	38

**Table 9 materials-15-08994-t009:** Point scores for criterion for aging tests (acc. to Table 4).

Configurations	Aging Test No. D7 and No. D11 acc. to the PN-EN ISO 9142 Standard				Sum
	R_cD7-2/D7-1_	R_cD7-3/D7-1_	R_cD11-2/D11-1_	R_cD11-3/D11-1_	
	Points				
Pine wood	5	4	5	5	19
Pine wood + X protection agent	4	3	5	5	17
Pine wood + Y protection agent	4	3	5	5	17

## Data Availability

Department of Wood Science and Thermal Techniques, Faculty of Forestry and Wood Technology, Poznan University of Life Sciences, Wojska Polskiego St. 38/42, 60-627 Poznan, Poland; tomasz.krystofiak@up.poznan.pl.

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
