# Peer review of "Adhesive Bonding of Scots Pine Wood from the Szczecinek Forest District for the Production of Garden Equipment: An Experimental Investigation"

_materials, 2022, doi:10.3390/ma15248994_

Round 1

Reviewer 1 Report

Dear authors, some comments.

The TITLE of the publication should be specified.

Abstract

Revision is required in light of the suggestions below for accurate use of terminology.

Introduction

The text contains some errors and inaccurate use of terms, such as:

Line 17 – ang

Line 35 – wood protection products do not require bactericidal properties

Line 37 – the term "eco-sanitary" is usually a medical term, "eco-safety" would be more appropriate.

Line 39 – “pressureless methods” – "non-pressure methods" are usually used for wood impregnation.

Lines 90, 96, 97, 332, 335 and elsewhere the authors use “wood protection substance”, “impregnating substance”, “preservative agents”, “preservation agent”, “tested substances” - “wood preservative” or “wood protection product” or “protection agent” would be more accurate.

In the text, it is recommended to use capital letters in the titles, e.g. Table, Test no. ...

Avoid using two terms in the text or explain the differences between them:

“Aging” or “Weathering” (standard PN-EN ISO 9142 uses the term "Aging")

“Strength” and “Durability”; “Strength” and “Resistance”

Clarifications are needed regarding the references, which in many places do not correspond to the text of the authors of the publication, for example:

Line 38 - references [2], [5], [9] do not correspond to the text "new generations of active substances, currently being developed with regard to eco-sanitary requirements", because they include both long-known (boron compounds) and currently limited preservatives (CCA).

Line 43, also line 111 – inappropriate reference to [10]. In this publication the authors do not consider glued products at all, stating that “Glued wood products (glulam, plywood, OSB, etc.) were excluded as their mechanical properties and permeability may be affected”.

Line 50 - [7] examines shear strength in friction welded joint, not wood bonded with glue, so it is not in context with the text about the required properties of the glue.

Line 60 – reference [19] did not study PUR adhesives at all.

Line 68 - [21] does not match the text.

Line 89 – [35] is not about gluing of impregnated wood at all.

Line 102 – the authors write “However, some investigations have found that impregnating substances do not have an adverse impact on the gluing process, and may even enable somewhat higher strength to be obtained”. Reference [2] is not about gluing impregnated wood at all; in the publication [48]: “All four tested fire-retardant systems and both fire-retardant retention levels were generally found to reduce plate shear and five-point flexural shear properties when compared to matched untreated plywood specimens”.  

Removing irrelevant references will only improve the publication.

Materials and Methods

Line 131 - dimensions of impregnated wood samples?

Line 133 – it is stated that preservative X contains "inorganic copper and boron compounds, and Cu-HDO copper-organic complex", but these are two different products. Specify!

Line 136 – The authors indicate that the wood is impregnated with the "pressure method", but no impregnation technological parameters are given; it is not specified what the concentrations of the working solutions were and what amounts of preservatives were introduced into the wood (preservative retention kg/m3). This is important, because the amount of preservative in the wood can significantly affect the gluing quality of the wood.

Line 139-141 – units for density, pressure, viscosity and time are given in square brackets.

Line 154 – clarify, were the test samples really obtained from trees of such different ages?

Line 162 – clarify the title of Table 3 "strength and resistance test"; the required shear strength values are given in the table.

Line 164-182 – values together with tolerances are usually put in parentheses, for example (23±2) °C instead of 23 (±2) °C.

Line 185 – indicate the strength testing machine Schopper ZDM 2.5/91 manufacturer (country, company).

Line 199 - 199 – numbering in Table 4 (1.; I., II., III.?)

Results

Line 213 – coefficient of variations designation "V" in the Tables 6-8 (correctly ν (ni) as elsewhere in the text).

Line 223 – “Figure 1. The course of glue lines strength depending on the adhesive bond surface for 223 untreated wood specimens after test no. 6”.  Figure shows data for wood treated with both preparations. What does "untreated specimens" mean? How was the adhesive bond surface determined? In the table, the designations of preservatives differ from those given in the text (X and Y). Figure 1 data is not discussed in the publication, so it is recommended to remove it.

Discussion

In the text, when discussing the results, it is necessary to refer to the data in the relevant tables (for example, Table 6; Table 7 and Table 8), and also indicate test D7 and/or D11.

Line 229 – the abbreviation WFP (wood failure percentage) is not explained; not specified how images of delamination were obtained (assessed visually?). Describe in the methodology!

Line 230 – ... “the value of the WFP indicator at destructive loads was 100%. This demonstrates that the weakest area of the tested joints was the glued wood” - What is the WFP indicator (also in the Line 358, Line 364) if wood failure is usually expressed as a percentage of the evaluated bonding surface? May be better “WFP value”?

Line 284 – The sentence "In the conditions of the procedure in including cyclic immersion of samples in water, freezing, drying and seasoning, it was found ..." should be clarified, because actually Test D7 according to the given methodology on page 5. included repeated soaking in water at 23ËšC and drying at the 55ËšC, but not freezing and seasoning (was shear strength determined for wet samples?). Insert a reference to Table 7 and Table 8 in the text.

Line 317 – “proposed evaluation criteria given in the table“ - which table?

Line 321 – “wood–preserver–adhesive” – better “preservative” or “protection agent”.

Line 327 - Table 9. “Scoring scores...”. Line 341 – Table 10. “Scoring for criterion I...” – better “Point scores...”. What is “criterion I”?

Recapitulation

Line 353 – “classification criteria” – in Table 4 and Table 5 the name is “assumpiton” criteria

Line 354 - ... ”PUR adhesive can be successfully used for the gluing” – it should be indicated that it applies to the studied adhesive

Line 355 - ... “high resistance to the effects of water, heat, and weathering” - what is meant here by weathering - cyclical aging according to D7 and D11?

Conclusions

Conclusions need to be revised (for example, it would be logical to combine p.3 and p. 4, p.2 and p.5).

Line 379 – “the joints made with PUR adhesive satisfied the 379 criteria for water resistance defined in the technical approval” - can it be claimed, considering that shear strength according to ITB technical approvals shear strength for conditioned wood under normal conditions is ≥9.0 MPa.

References

Publications in Polish and Russian should also be given titles in English.

There are inaccuracies in the titles of the publications (also in the titles in German), so they should be revised, especially 1, 2, 4, 6, 8, 10, 12, 17, 20, 22, 29, 44, 51, 62.

Author Response

Dear Editor and Reviewers,

First of all we would like to thank the Editor and Reviewers for their kind words and useful comments about our paper. We have now carefully addressed all points in question. The changes to the main text are marked yellow. A point-by-point response to the raised questions is shown below:

If you have additional comments regarding the revised manuscript, please let us know.

  1. The TITLE of the publication should be specified.

Answer:

The title of the paper was specified.

  1. Abstract

Revision is required in light of the suggestions below for accurate use of terminology.

Answer:

Accordingly to the reviewer suggestions changes in the terminology were made.

  1. Introduction

The text contains some errors and inaccurate use of terms, such as:

Answer:

Errors and inaccurate use of terms were corrected.

Line 17 – ang

Line 35 – wood protection products do not require bactericidal properties

"Bactericidal properties" was removed.

Line 37 – the term "eco-sanitary" is usually a medical term, "eco-safety" would be more appropriate.

Changed "eco-sanitary" into "eco-safety".

Line 39 – “pressureless methods” – "non-pressure methods" are usually used for wood impregnation.

Changed to surface methods (lubrication, spraying and soaking) were added.

Lines 90, 96, 97, 332, 335 and elsewhere the authors use “wood protection substance”, “impregnating substance”, “preservative agents”, “preservation agent”, “tested substances” - “wood preservative” or “wood protection product” or “protection agent” would be more accurate.

Thank you very much for your kind advice. Changed “wood protection substance” into “wood protection agent”and „protection agent“.

In the text, it is recommended to use capital letters in the titles, e.g. Table, Test no. ...

Capital letters in the titles were used.

  1. Avoid using two terms in the text or explain the differences between them:

“Aging” or “Weathering” (standard PN-EN ISO 9142 uses the term "Aging")

“Strength” and “Durability”; “Strength” and “Resistance”

Answer:

Thank you very much for your kind advice. We strongly agree with the reviewer. The term „aging“ (accordingly to the PN-EN ISO 9142 standard) was used.

By using the term "strength", the authors meant the strength of the joints after test No. 1 acc. to the AT-15-2948/00 (samples after gluing were conditioned for 7 days 20 ± 2 °C and RH 65 ± 5%). However "durability" - the strength registered after simulation of conditions (tests acc. to the AT-15-2948/00 procedure): water resistance No. 2, 3 and thermoresistance No. 6, and aging tests D7 and D11).

Clarifications are needed regarding the references, which in many places do not correspond to the text of the authors of the publication, for example:

  1. Line 38 - references [2], [5], [9] do not correspond to the text "new generations of active substances, currently being developed with regard to eco-sanitary requirements", because they include both long-known (boron compounds) and currently limited preservatives (CCA).

Line 43, also line 111 – inappropriate reference to [10]. In this publication the authors do not consider glued products at all, stating that “Glued wood products (glulam, plywood, OSB, etc.) were excluded as their mechanical properties and permeability may be affected”.

Line 50 - [7] examines shear strength in friction welded joint, not wood bonded with glue, so it is not in context with the text about the required properties of the glue.

Line 60 – reference [19] did not study PUR adhesives at all.

Line 68 - [21] does not match the text.

Line 89 – [35] is not about gluing of impregnated wood at all.

Line 102 – the authors write “However, some investigations have found that impregnating substances do not have an adverse impact on the gluing process, and may even enable somewhat higher strength to be obtained”. Reference [2] is not about gluing impregnated wood at all; in the publication [48]: “All four tested fire-retardant systems and both fire-retardant retention levels were generally found to reduce plate shear and five-point flexural shear properties when compared to matched untreated plywood specimens”.  

Removing irrelevant references will only improve the publication.

Answer:

Thank you very much. We agree with the reviewer. All irrelevant references were removed.

  1. Materials and Methods

Line 131 - dimensions of impregnated wood samples?

Answer:

Dimensions of samples were given to the text.

  1. Line 133 – it is stated that preservative X contains "inorganic copper and boron compounds, and Cu-HDO copper-organic complex", but these are two different products. Specify!

Answer:

Thank you the reviewer for the suggestion. Changes have been made to the article text.

  1. Line 136 – The authors indicate that the wood is impregnated with the "pressure method", but no impregnation technological parameters are given; it is not specified what the concentrations of the working solutions were and what amounts of preservatives were introduced into the wood (preservative retention kg/m3). This is important, because the amount of preservative in the wood can significantly affect the gluing quality of the wood.

Answer:

Thank you the reviewer for the suggestion. Both protection agents were in the 4 class. Samples for experiments were prepared in the industrial conditions. Due to the confidentiality clause, details of the impregnation have been omitted. In the next investigations we will analyse influence of the preservative retention upon strength of glue lines.

  1. Line 139-141 – units for density, pressure, viscosity and time are given in square brackets.

Answer:

Removed square brackets

  1. Line 154 – clarify, were the test samples really obtained from trees of such different ages?

Answer:

The experimental material was obtained from the Forest District of the Regional Directorate of State Forests (RDSF) in Szczecinek. The authors, however, do not have data on the age of the raw material, which was intended for impregnation and later gluing. Elements were taken straight from production. In industrial practice, such selection is not carried out, only uses a diverse material.  The Table has been deleted.

  1. Line 162 – clarify the title of Table 3 "strength and resistance test"; the required shear strength values are given in the table.

Answer:

The authors previously explained the differences between strength and durability. Table 3 contains both the test conditions and the strength values that the samples should achieve to meet the requirements (acc. to the ITB technical approval).

  1. Line 164-182 – values together with tolerances are usually put in parentheses, for example (23±2) °C instead of 23 (±2) °C.

Answer:

We agree with the reviewer. All values together with tolerances were put in parentheses.

  1. Line 185 – indicate the strength testing machine Schopper ZDM 2.5/91 manufacturer (country, company).

Answer:

Information related to the Schopper testing maschine was added.

  1. Line 199 - 199 – numbering in Table 4 (1.; I., II., III.?)

Answer:

The numbering in the Table 4 concerns to the formula number. A mistake was made during the numbering. Numbering has been corrected.

Results

  1. Line 213 – coefficient of variations designation "V" in the Tables 6-8(correctly ν (ni) as elsewhere in the text).

Answer:

The error has been corrected.

  1. Line 223 – “Figure 1. The course of glue lines strength depending on the adhesive bond surface for 223 untreated wood specimens after test no. 6”.  Figure shows data for wood treated with both preparations. What does "untreated specimens" mean? How was the adhesive bond surface determined? In the table, the designations of preservatives differ from those given in the text (X and Y).Figure 1 data is not discussed in the publication, so it is recommended to remove it.

Answer:

We agree with the reviewer. The Figure 1 was removed.

Discussion

  1. In the text, when discussing the results, it is necessary to refer to the data in the relevant tables (for example, Table 6; Table 7 and Table 8), and also indicate test D7 and/or D11.

Answer:

Referred to data in the relevant tables.

  1. Line 229 – the abbreviation WFP (wood failure percentage) is not explained; not specified how images of delamination were obtained (assessed visually?). Describe in the methodology!

Answer:

The abbreviation WFP (wood failure percentage) in the methodology section was explained.

  1. Line 230 – ... “the value of the WFP indicator at destructive loads was 100%. This demonstrates that the weakest area of the tested joints was the glued wood” - What is the WFPindicator (also in the Line 358, Line 364) if wood failure is usually expressed as a percentage of the evaluated bonding surface? May be better “WFP value”?

Answer:

Corrected for WFP value and in the methodology added description of the conducting assessment.

  1. Line 284 – The sentence "In the conditions of the procedure in including cyclic immersion of samples in water, freezing, drying and seasoning, it was found ..." should be clarified, because actually Test D7 according to the given methodology on page 5. included repeated soaking in water at 23ËšC and drying at the 55ËšC, but not freezing and seasoning (was shear strength determined for wet samples?). Insert a reference to Table 7 and Table 8 in the text.

Answer:

Thank you very much the reviewer. The words “freezing” and “seasoning” were removed.

  1. Line 317 – “proposed evaluation criteria given in the table“ - which table?

Answer:

We added table number (Table 9).

  1. Line 321 – “wood–preserver–adhesive” – better “preservative” or “protection agent”.

Answer:

Changed into “protection agent”.

  1. Line 327 - Table 9. “Scoring scores...”.

Answer:

Changed into: “Point scores”.

  1. Line 341 – Table 10. “Scoring for criterion I...” – better “Point scores...”. What is “criterion I”?

Answer:

As criterion 1, the authors mean the calculation formula in the Table 5. Removed for criterion I (there is only one formula).

Recapitulation

  1. Line 353 – “classification criteria” – in Table 4 and Table 5 the name is “assumpiton” criteria

Answer:

Changed “classification” into “assumption”.

  1. Line 354 - ... ”PUR adhesive can be successfully used for the gluing” – it should be indicated that it applies to the studied adhesive

Answer:

It can be stated that the PUR adhesive used in the tested joints can be successfully used for gluing of pine wood protected with the substances considered in the work, obtaining joints with high durability to the of water, heating and aging.

  1. Line 355 - ... “high resistance to the effects of water, heat, and weathering” - what is meant here by weathering - cyclical aging according to D7 and D11?

Answer:

Atmospheric factors - factors that cause aging processes in this case cyclic aging according to the D7 and D11 tests. This information was given earlier in the Materials and methods section. 

Conclusions

  1. Conclusions need to be revised (for example, it would be logical to combine p.3 and p. 4, p.2 and p.5).

Answer:

Conclusions No. 3 with No. 4 and No. 2 with No. 5 were combined.

  1. Line 379 – “the joints made with PUR adhesive satisfied the criteria for water resistance defined in the technical approval” - can it be claimed, considering that shear strength according to ITB technical approvals shear strength for conditioned wood under normal conditions is ≥9.0 MPa.

Answer:

The average values obtained in the measurements were in some cases at a level lower than that specified in the ITB approval. Evaluation of strength in relation to the maximum values, indicates for all systems to exceed them. In the case of water resistance tests, test No. 2 and 3 and thermal resistance test No. 6 were met for impregnated samples also in average values. The carried out evaluation of the delamination mechanisms of the adhesive joints proved that the weakest zone of the evaluated joints was the glued wood.

References

  1. Publications in Polish and Russian should also be given titles in English.

There are inaccuracies in the titles of the publications (also in the titles in German), so they should be revised, especially 1, 2, 4, 6, 8, 10, 12, 17, 20, 22, 29, 44, 51, 62.

Answer:

Accordingly to the reviewer suggestions changes were made to the text.

At this point, we would like to once again thank the Dear Reviewer, for the substantive comments, which helped us to improve this article.

Reviewer 2 Report

Line 69. Glued laminated timbers and plywood are routinely pressure treated after layup. And treating cylinders are pretty big – I doubt there is much gardening equipment that couldn’t fit in one!

The number of drying cycles is precisely the same whether the impregnation is before or after the layup. A big disadvantage of pre-layup treatment is the generation of treated wood residues when planing the material in preparation for bonding. Plus, with refractory species (not pine sapwood), you risk removing much of the treated wood ‘envelope’ that protects the wood.

L138 and details on the preservative process and results? Retention and penetration data?

L139 were the lamellae planed before bonding? If so, how much of the preservative treatment was lost?

Line 153 Are the table 1 relevant in any way?

L221 Figure 1… is this useful? It doesn’t show up in any of the results or discussions. Seems like it isn’t needed.

L238 and following. I don’t think you need to explain and discuss the reductions in properties associated with, for example, wetting. Its common knowledge that wet wood is weaker. Likewise, the other components of the standard test methods (wetting and drying) are known to stress the bond – that’s why they in in the test. For example line 29-297 really aren’t needed. The only thing to need to discuss really is the potential differences among the treatment groups.

Discussion should include discussion of the potential disadvantages of pre-bonding treatment.

Author Response

Dear Editor and Reviewers,

First of all we would like to thank the Editor and Reviewers for their kind words and useful comments about our paper. We have now carefully addressed all points in question. The changes to the main text are marked yellow. A point-by-point response to the raised questions is shown below:

If you have additional comments regarding the revised manuscript, please let us know.

Line 69. Glued laminated timbers and plywood are routinely pressure treated after layup. And treating cylinders are pretty big – I doubt there is much gardening equipment that couldn’t fit in one!

Answer:

The number of drying cycles is precisely the same whether the impregnation is before or after the layup. A big disadvantage of pre-layup treatment is the generation of treated wood residues when planing the material in preparation for bonding. Plus, with refractory species (not pine sapwood), you risk removing much of the treated wood ‘envelope’ that protects the wood.

L138 and details on the preservative process and results? Retention and penetration data?

Answer:

Thank you the reviewer for the suggestion. Both protection agents were in the 4 class. Samples for experiments were prepared in the industrial conditions. Due to a confidentiality clause, details of the impregnation have been omitted. In the next experiments we will analyse preservative retention.

L139 were the lamellae planed before bonding? If so, how much of the preservative treatment was lost?

Answer:

No, the lamaellae were not planed before bonding.

Line 153 Are the table 1 relevant in any way?

Answer:

The table shows the taxation of the forest in Szczecinek - the region from which the experimental material came. The authors do not have data on the age of the raw material that was intended for the study. In industrial practice, no such selection is carried out, but a diverse material is used. It was decided to remove it.

L221 Figure 1… is this useful? It doesn’t show up in any of the results or discussions. Seems like it isn’t needed.

Answer:

Considering that Fig. 1 shows the formation of adhesive strength depending on the glue-lines area only for test No. 6. We agree with the reviewer's suggestion that it is unnecessary.  The figure has been removed.

L238 and following. I don’t think you need to explain and discuss the reductions in properties associated with, for example, wetting. Its common knowledge that wet wood is weaker. Likewise, the other components of the standard test methods (wetting and drying) are known to stress the bond – that’s why they in in the test. For example line 29-297 really aren’t needed. The only thing to need to discuss really is the potential differences among the treatment groups.

Answer:

The aim of the work was to determine the gluability of pine wood impregnated with two protection agent used in production technologies for garden equipment, and their effect on the strength, water resistance and thermal resistance of joints, as well as susceptibility to aging. We wrote in the paper „Analysis of the strength of the joints compared with the maximum values showed, unambiguously for all systems, that the PUR adhesive used, applied both to impregnated and unprotected wood, provides the technical possibility of obtaining water-resistant glued joints“.

The authors' intention was to quantify the changes in strength after the resistance tests. For this purpose, a point scale was used. The authors propose to leave this section in the text.

The potential differences among the treatment groups will be the aim of the next paper.

Discussion should include discussion of the potential disadvantages of pre-bonding treatment.

Answer:

Information was added to the text.

Round 2

Reviewer 1 Report

No.

Author Response

Dear Reviewer,

Thank you very much for your very kind reply. In accordance with your suggestions, we have made changes and additions to the text. Your comments will be very useful to us in the preparation of future papers.

yours sincerely,

Tomasz Krystofiak